# A Unified Nonlinear Elastic Model for Rock Material

Chong Chen [1,2], Shenghong Chen [1,*], Yihu Zhang [2], Hang Lin [3] and Yixian Wang [4]

1. School of Water Resources and Hydropower Engineering, Wuhan University, Wuhan 430072, China
2. Changjiang River Scientific Research Institute, Wuhan 430010, China
3. School of Resources and Safety Engineering, Central South University, Changsha 410083, China
4. College of Civil Engineering, Hefei University of Technology, Hefei 230009, China
* Correspondence: chensh@whu.edu.cn

**Abstract:** Under conditions of low or medium stress, rocks that are in the compression state exhibit perceivable nonlinear elastic characteristics. After a comprehensive review of the existing nonlinear elastic models of rocks and joints, we proposed a new unified nonlinear elastic model. This new model contains two parameters with clear definitions, namely, the initial elastic modulus $E_{ni}$ and the modulus change rate $m$. This model covers a variety of existing models, including the simple exponential model, BB model and two-part Hooke's model, etc. Based on the RMT experimental system, a large number of uniaxial compression tests for dolomite, granite, limestone and sandstone were performed, and their nonlinear deformation stress-strain curves were obtained and fit with the unified nonlinear elastic model. The results show that the rocks have obvious nonlinear elastic characteristics in their initial compression stage, and that the unified nonlinear elastic model is able to describe these phenomena rather well. In addition, an empirical formula for predicting the uniaxial compressive strength of the rock was constructed, corresponding to the unified nonlinear elastic model.

**Keywords:** rock; deformation; linear elastic model; nonlinear elastic model

## 1. Introduction

Engineering rock masses are discontinuous and often heterogeneous and anisotropic. Jointed rock masses are the basic units that are used to study the properties of engineering rock masses, and their deformation characteristics under different stress conditions are crucial to the performance of geotechnical engineering structures. Since a large number of projects are located in the shallow layer of Earth, which experiences low crustal stress, it is of great significance to study the deformation characteristics of jointed rock masses under conditions of low and medium stress smaller than 10 MPa and 20 MPa, respectively [1]. Under such circumstances, both intact rocks and joints (as two components of the jointed rock mass) will undergo nonlinear elastic deformation.

Many scholars have performed extensive research on the nonlinear deformation of joints and different models have been proposed. Shehata [2] first used semilogarithmic functions in order to describe the normal deformation characteristics of joints. Goodman [3] used the hyperbolic model in order to describe the relationship between normal stress and the normal closure of joints. On this basis, Bandis and Barton [4,5] proposed a relatively simple model by introducing two parameters: initial joint stiffness and maximum closure. Malama and Kulatilake [6] proposed a simple exponential model and a modified exponential model. Swan and Sun [7,8] proposed a power function model based on Hertzian contact theory. Other scholars [9–11] have also developed and improved the above models and the nonlinear elastic deformation of joints has had a great influence on their shear strength [12,13]. Since it is difficult to directly measure the deformation of joints, it is indirectly evaluated by subtracting the deformation of the intact rock specimen from the jointed rock mass. However, during such an operation the nonlinear elastic deformation property of the intact rock is unconcernedly ignored (although not always).

The nonlinear elastic deformation in the initial compaction stage of the intact rock was customarily considered to be caused by sampling disturbance [14]. Without considering the initial compaction stage, hyperbolic [15], logarithmic [16], double exponential [17] and exponential [18] stress–strain models for different rock types have been proposed. However, there is a significant difference in the nonlinear part of stress–strain relationships between the observed and estimated values [19]. The widely used stress–strain curves for intact rocks with cracks [20] and weak rocks [21] have validated that the nonlinear elastic deformation in the initial compaction stage is remarkable. It has been shown that the porosity of the rock and the fractures in the rock can not only reduce its strength and elastic modulus [22,23], but also cause its nonlinear characteristics [24]. Jiang [25] and Zhang [26] proposed different improved models in order to describe the rock compression stage based on the Duncan model. Wang [27] introduced a compression factor in order to characterize the change in porosity and established a nonlinear damage constitutive model reflecting the rock compression stage. Based on particle contact theory, Ma [28] established a constitutive model by using spring elements and G-W contact elements in order to express the linear and nonlinear deformation of contacts between adjacent particles, respectively. Liu [29,30] used engineering strains and natural strains based on Hooke's law to describe the deformation of the rock matrix and the porosity and fractures respectively, and proposed the two-part Hooke's model. This model was used to quantitatively calculate the crack closure stress of the rock [31,32] and was introduced into the statistical damage constitutive model of the rock [33,34]. Using numerical computation and the distinct element method, it was proven that the porosity of the rock or the microfractures that are present in the rock can bring about nonlinear elastic deformation in the initial compaction stage of middle to hard rock [35]. However, this still needs to be verified using appropriate mathematical models and experiments.

In this paper, the ability of existing nonlinear elastic models for rocks and/or joints was discussed and analyzed. Then, a unified nonlinear elastic model was proposed. Based on a large number of experiments investigating the mechanics of intact rocks, the nonlinear elastic deformation of middle to hard rocks was analyzed. Afterwards, comparisons were made between the parameters of the unified nonlinear elastic model and the experimental data. In this manner, the validity of our model was justified.

## 2. Common Nonlinear Elastic Models

### 2.1. Nonlinear Elastic Models of Joints

Goodmam [3] proposed a hyperbolic function in order to describe the deformation of joints based on their initial normal stress and maximum allowable closure.

$$\sigma_n = [d_n/(d_{\max} - d_n)]\sigma_0 + \sigma_0 \tag{1}$$

Here, $d_{max}$ is the maximum allowable closure, $\sigma_0$ is the initial normal stress, $\sigma_n$ is the normal stress and $d_n$ is the normal closure.

Bandis and Barton [4,5] conducted a large number of experiments on joints with different weathering degrees and types of rocks. By introducing the initial stiffness $K_{ni}$ as an important parameter for joints, the classical BB model was established based on the Goodman's hyperbolic model. The formula of the BB model is as follows:

$$d_n = \sigma_n/[K_{ni} + (\sigma_n/d_{\max})] \tag{2}$$

The above equation can be rewritten as

$$\sigma_n = \frac{K_{ni}d_{\max}}{d_{\max}/d_n - 1} \tag{3}$$

Malama and Kulatilake [6] proposed a simple exponential model with reference to the BB model. The formula of this simple exponential model is as follows:

$$d_n = d_{\max}\left[1 - \exp\left(-\frac{\sigma_n}{K_{ni}d_{\max}}\right)\right] \tag{4}$$

The above equation can be rewritten as

$$\sigma_n = K_{ni}d_{\max}\ln\left(\frac{1}{1 - d_n/d_{\max}}\right) \tag{5}$$

Malama and Kulatilake [6] found that under medium stress, the normal closure calculated by the hyperbolic model was slower than that calculated using the experimental results, whereas the simple exponential model was faster. By introducing the half value $\sigma_{1/2}$ as a new parameter, which is the stress when the joint closure reached half of the maximum allowable closure, the modified exponential model was proposed. The modified exponential model excludes initial joint stiffness $K_{ni}$ in its formula, which is an important parameter [10], and its physical meaning is not clear.

The formula of the modified exponential model is as follows:

$$d_n = d_{\max}\left\{1 - \exp\left[-\left(\frac{\sigma_n}{\sigma_{1/2}}\right)^n \ln 2\right]\right\} \tag{6}$$

### 2.2. Nonlinear Elastic Models of Joints

The typical stress-strain curve of the rock specimen under uniaxial stress is shown in Figure 1, which can be divided into four stages: (1) the compression of pores and fractures (section OA); (2) elastic deformation and the stable development of microfractures (section AC), where point B is the elastic limit; (3) the unstable developmental stage of the fractures (section CD); and (4) the stage after rock rupture (after point D). As shown in Figure 1, the deformation rate slows down with increasing stress in the compression stage (section OA), which is because the elastic modulus increases with decreasing rock specimen length. In order to distinguish between the nonlinear elastic and linear elastic deformation of the rock in Figure 1, a straight line (OB' parallel to AB) through the coordinate origin was drawn, and the difference between the curve OAB and the straight line OB' was the curve OA', as shown in Figure 2. The straight line OB' and the curve OA' represent the linear elastic and the nonlinear elastic deformation of the rock, respectively.

As shown in Figure 2, both linear elastic deformation and nonlinear elastic deformation will occur when the strain is below point A. Nonlinear elastic deformation will terminate after reaching its maximum allowable deformation when the strain reaches point A, and only linear elastic deformation will occur when the strain exceeds it. The stress corresponding to point A is called the closing stress of the initial microcrack, which is also one of the important parameters of the rock.

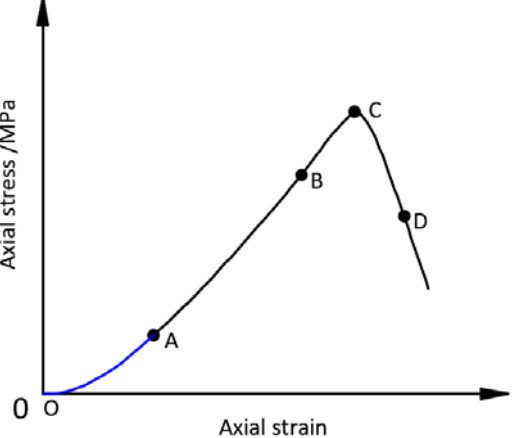

**Figure 1.** The stress-strain curve of a uniaxial compression test adapted from Corkum. Adapted from [21].

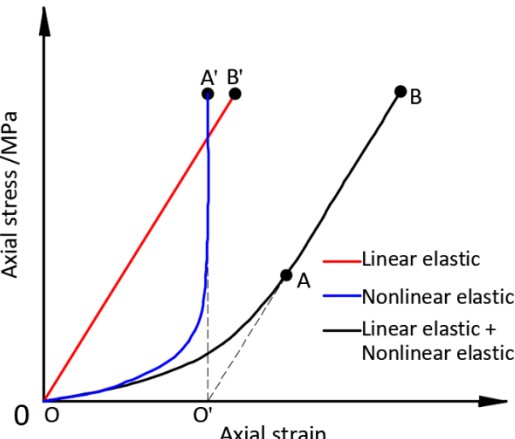

**Figure 2.** Linear and nonlinear elastic deformation of rock adapted from Liu. Adapted from [30].

Since a rock is not an ideal linear elastic material and is a heterogeneous material composed of solid skeletons with pores and fractures filling the space between them, Liu [29,30] divided the rock into hard and soft parts, which have linear elastic and nonlinear elastic deformation characteristics, respectively. The two-part Hooke's model [29,30] formula is as follows:

$$\varepsilon = \gamma_e \frac{\sigma}{E_e} + \gamma_t \left[ 1 - \exp\left( \frac{-\sigma}{E_t} \right) \right] \tag{7}$$

where $E_e$ and $E_t$ are the elastic modulus of the hard and soft parts, respectively, and $\gamma_e$ and $\gamma_t$ are the proportions of the hard and soft parts of the rock, respectively. This model can be used for composite materials [36], such as material combined with coal and rock [37].

If the deformation of the soft part to its initial length is $\varepsilon_t$, the nonlinear elastic deformation of the soft part in the two-part Hooke's model can be expressed as

$$\sigma = E_t \ln\left( \frac{1}{1 - \varepsilon_t} \right) \tag{8}$$

## 3. A New Unified Nonlinear Elastic Model

This section presents a new unified nonlinear elastic model for rock material based on Hooke's law. Hooke's law can be used to describe the linear relationship between the strain and stress of linear elastic material under compression. When the compression direction is positive, Hooke's law is expressed as

$$\sigma = E\varepsilon \tag{9}$$

$$d\sigma = Ed\varepsilon \tag{10}$$

$$\varepsilon = -\frac{L - L_0}{L_0} \tag{11}$$

$$d\varepsilon = -\frac{dL}{L_0} \tag{12}$$

where $E$, $\varepsilon$ and $L$ are the elastic modulus, strain and length of the material under stress $\sigma$, respectively, and $L_0$ is the initial length of the material.

The elastic modulus of the linear elastic material remains unchanged during deformation, as shown in Figure 3a. The $E_{ni}$ parameter is introduced, which represents the initial elastic modulus of the material, and we have the following:

$$E = E_{ni} \tag{13}$$

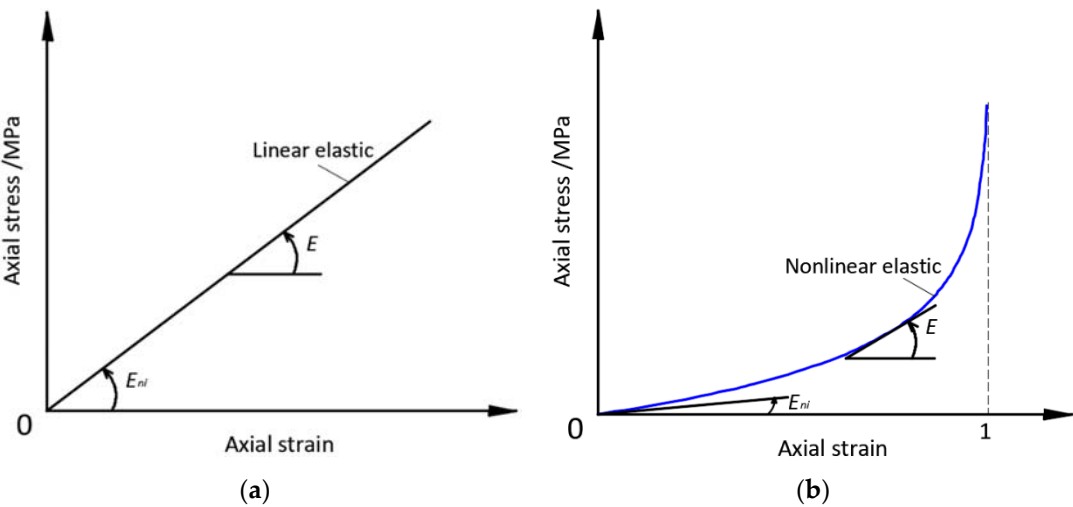

**Figure 3.** Linear and nonlinear elastic curves (**a**-Linear elastic; **b**-Nonlinear elastic).

Substituting Equations (12) and (13) into (10) yields the following:

$$d\sigma = -E\frac{dL}{L_0} = -E_{ni}\frac{dL}{L_0} \tag{14}$$

If the elastic modulus of the material is not a fixed value, but is related to the material length during deformation, its stress-strain curve is nonlinear, as shown in Figure 3b.

The elastic modulus during deformation follows the following formula:

$$E = E_{ni}\left(\frac{L_0}{L}\right)^m \quad (m \geq 0) \tag{15}$$

where $m$ is a constant.

The nonlinear elastic curve has obvious differences compared to the linear elastic curve. In the nonlinear elastic curve, the closer to the maximum strain, the faster the curve rises, and more stress is required for further deformation of the material. As shown by Equation (15), the elastic modulus of the material decreases with increasing length during the deformation process.

Substituting Equations (12) and (15) into (10) yields the following:

$$d\sigma = -E_{ni}L_0^{m-1}\frac{dL}{L^m} \tag{16}$$

The length $L$ of the material under stress $\sigma$ can be obtained by integrating Equation (16). The $m$ value is a special case when it is equal to 1, and the derivation method is different from that when it is not equal to 1.

When $m$ is equal to 1

$$L = L_0 \exp\left(-\frac{\sigma}{E_n}\right) \tag{17}$$

When $m$ is not equal to 1

$$L = L_0\left(\frac{\sigma(m-1)}{E_{ni}} + 1\right)^{\frac{1}{1-m}} \tag{18}$$

After substituting Equations (17) and (18) into (11) and simplifying the formula, we have the following:

When $m$ is equal to 1

$$\sigma = E_{ni}\ln\left(\frac{1}{1-\varepsilon}\right) \tag{19}$$

When $m$ is not equal to 1

$$\sigma = E_{ni}\frac{(1-\varepsilon)^{1-m}-1}{m-1} \tag{20}$$

The new unified nonlinear elastic model is composed of Equations (19) and (20). Special forms of the formula exist when $m$ is equal to 0, 1 and 2.

When $m$ is equal to 0, the unified nonlinear elastic model can be simplified into a linear elastic model:

$$\sigma = E_{ni}\varepsilon \tag{21}$$

When $m$ is equal to 1, the unified nonlinear elastic model is Equation (19), it is the same as Equation (8) of the two-part Hooke's model, and it is the same as Equation (5) of the simple exponential model under the following conditions:

$$\begin{cases} E_{ni} = K_{ni}d_{\max} \\ \varepsilon = d_n/d_{\max} \end{cases} \tag{22}$$

When $m$ is equal to 2, Equation (20) can be simplified as

$$\sigma = E_{ni}\frac{1}{1/\varepsilon - 1} \tag{23}$$

It is the same as Equation (3) of the BB model when Equation (22) is established.

Note that the stress displacement coordinate system is often used for the nonlinear deformation of joints, and if it is changed to the stress–strain coordinate system that is the same as the rock, Equation (22) will be established.

In addition, when the unified nonlinear elastic model is used for the rock, its strain is the deformation divided by the allowable nonlinear deformation, that is, the two coordinate systems in Figures 2 and 3b are different. Because the allowable nonlinear deformation of the rocks in the initial compaction stage in Figure 3b is generally less than 0.5%, the differences between the elastic modulus values calculated by the two coordinate systems are also less than 0.5%.

In general, the unified nonlinear elastic model has two key parameters, $E_{ni}$ and $m$, where $E_{ni}$ is the initial elastic modulus and $m$ represents its change rate, and both have reasonable physical significance. The unified nonlinear elastic model can include the existing main nonlinear elastic models of the rocks and joints and can also express some new models.

## 4. The Characteristics of the New Model

### 4.1. Sensitivity Analysis of the Parameters

The new unified nonlinear elastic model has two parameters: $E_{ni}$ and $m$. The influence of the $E_{ni}$ values was studied by fixing the $m$ value to 1.0. The stress-strain curves of the new model when $E_{ni}$ is equal to 1 MPa, 5 MPa, 10 MPa, 15 MPa and 20 MPa are listed in Figure 4. All the curves rise linearly when the strain is less than 0.4, and nonlinearity gradually appears when the strain is greater than 0.4. The higher the $E_{ni}$ values are, the higher the position of the curves, and the distance between adjacent curves is equal when the strain is the same. The slope of the curve for the unified nonlinear elastic model in its initial stage is determined by the $E_{ni}$ value.

The influence of the $m$ value was studied by fixing the $E_{ni}$ value to 10. The stress-strain curves of the new model when the $m$ value is equal to 0, 0.5, 1.0, 1.5 and 2.0 are listed in Figure 5. All the curves are coincident when the strain is less than 0.4, and differences appear quickly when the strain is greater than 0.4. The curvature of the curves for the unified nonlinear elastic model in its middle–late stages is determined by the $m$ value.

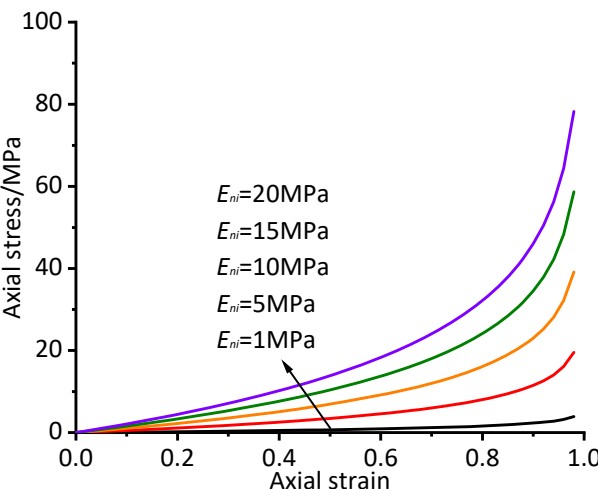

**Figure 4.** Stress-strain curves for different $E_{ni}$ values ($m$ = 1.0).

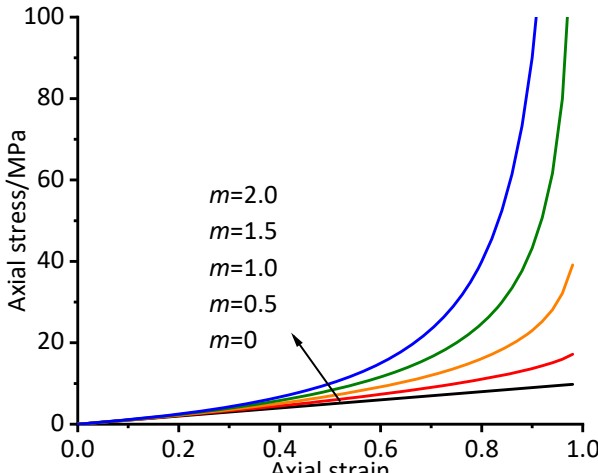

**Figure 5.** Stress-strain curves for different $m$ values ($E_{ni}$ = 10 MPa).

Compared with Figures 4 and 5, the mechanisms of influence for the two parameters of the unified nonlinear elastic model on the stress-strain curves are different since they represent different physical properties of the nonlinear elastic deformation of the rock. The initial stage of the curve for the unified nonlinear elastic model is determined by the $E_{ni}$ value, and its middle–late stages are determined by the $m$ value. The unified nonlinear elastic model uses the above two parameters to jointly adjust and control in order to achieve accurate descriptions of the nonlinear elasticity of rock.

In fact, the nonlinear deformation of the rock or joint is not obvious when the deformation is small. During the closure of the rock pores and fractures or the joint, the anti-deformation ability increases as the rock matrix gradually contacts, and then the nonlinear elastic characteristic gradually becomes obvious. The new unified nonlinear elastic model can accurately describe these properties.

*4.2. Approximating to a Single Stress-Strain Curve*

The initial compression stage of a rock specimen represented by the single stress-strain curve was calculated using the method shown in Figure 1. For different ranges of stress-strain curves with strain from 0 to 0.5, 0.6, 0.7, 0.8, 0.9 and 1.0, the unified nonlinear elastic model was used to fit them using the least square algorithm method. The values of $E_{ni}$ and $m$ for the unified nonlinear elastic model are obtained, and the correlation coefficient $R^2$ values were recorded. The horizontal axis in Figure 6 represents the different ranges of the stress-strain curve, with the $E_{ni}$, the $m$ values and the $R^2$ values located on different vertical

axes. The red line represents the $R^2$ values, and its fluctuation range is very small. All the $R^2$ values are greater than 0.98, which showed a good relevant fit for the different ranges. The values of $E_{ni}$ decrease with a larger range, whereas the $m$ value increases with it. These trends become obvious when the strain is greater than 0.7 and very obvious when it is greater than 0.95. There is a negative correlation between the $E_{ni}$ values and the $m$ values.

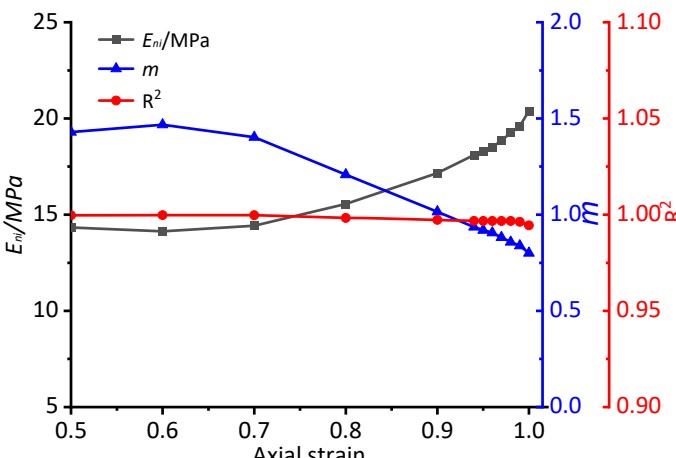

**Figure 6.** Fitting results of different strain ranges.

The different curves of the unified nonlinear elastic model with the $E_{ni}$ values and $m$ values, which were both obtained after fitting different curve ranges, are listed in Figure 7. As the fitting curve range increases, the calculated curve gradually approaches the target curve. The red line was calculated using the parameters after fitting the curve from 0 to 1.0, and is the curve of best fit.

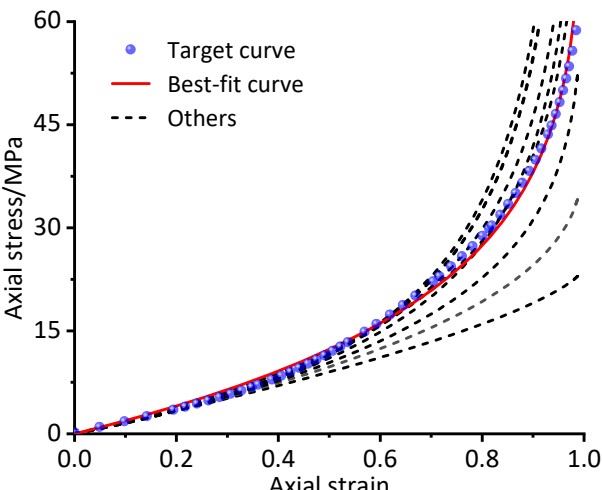

**Figure 7.** Approximating to a single stress-strain curve.

## 5. Experimental Validation of the New Model

### 5.1. Experimental Methods

The fitting effect of the unified nonlinear elastic model and whether it could express the nonlinear elastic deformation of different rock types required verification by indoor experiments. Four different types of middle to hard rocks were selected for uniaxial compression testing with the rock mechanics test system. The four types of rock were dolomite, granite, limestone and sandstone. Cylindrical samples were prepared according to the suggested methods proposed by the International Society of Rock Mechanics and Rock Engineering (IRSM) [38]. The samples were 50 mm in diameter and 100 mm in length. The appearance of the samples was checked to ensure that they had no foliation,

joints or big fractures and that, to some extent, they were intact rock specimens. The differences between the four rock types were directly attributed to the mineral components, porosity and microfractures. The samples of the four rock types have the same brittle failure characteristics.

The tests were performed with a loading speed of 0.05 m/s in the axial direction and with no confining pressure. The displacement sensors were installed at the end and side surface of the rock specimens in order to measure their deformation, and their precision was not higher than 0.01 mm. The rock mechanics test system and the rock specimens are shown in Figures 8 and 9, respectively. During the experiments, the uniaxial stress-strain curves were recorded as listed in Figure 10. According to the values of the uniaxial compressive strength, the stress-strain curves of the ten rock specimens from every rock type were selected for analysis.

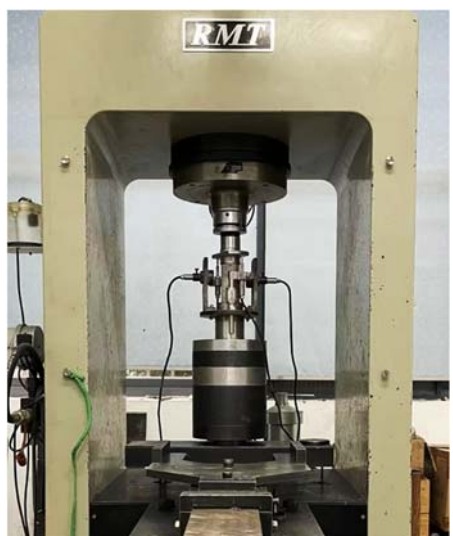

**Figure 8.** Rock mechanics test system (RMT).

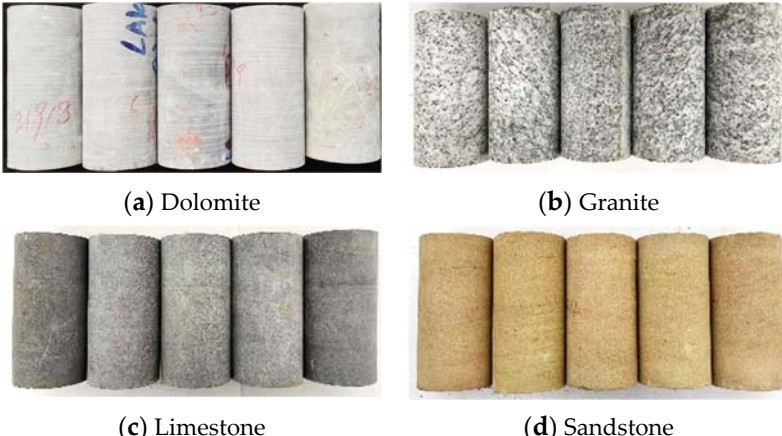

| | |
|:---:|:---:|
| (**a**) Dolomite | (**b**) Granite |
| (**c**) Limestone | (**d**) Sandstone |

**Figure 9.** Different rock specimens.

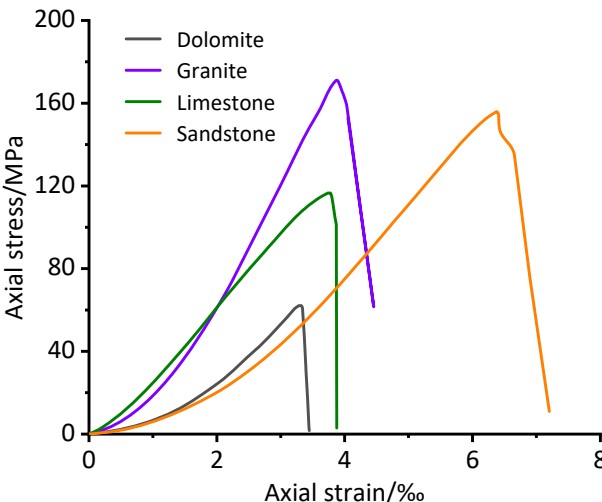

**Figure 10.** Typical stress-strain curves.

### 5.2. Experimental and Fitting Results

Based on the stress-strain curves under uniaxial compression, the uniaxial compressive strength was obtained, and the stress-strain curves of all the rock specimens in the initial compression stage were calculated using the method shown in Figure 1. The unified nonlinear elastic model was used to fit the curves using the least square algorithm method. The $E_{ni}$ and $m$ values that were obtained by fitting the curves and the coefficients of correlation $R^2$ values are listed in Table 1. This new model shows a better goodness of fit and a higher accuracy of fit since all the $R^2$ values of the specimens exceed 0.97. The $UCS$ values, the $E_{ni}$ values and the $m$ values of each rock specimen were different. The $E_{ni}$ values increased as the $UCS$ value increased for each rock type, whereas the $m$ values changed slightly.

The nonlinear elastic deformation of the rocks in the initial compression stage exists objectively, and their regularity can be followed as well. The unified nonlinear elastic model is suitable for describing the nonlinear elastic deformation behavior of intact rocks at the initial compression stage.

**Table 1.** Experimental and fitting results (D—Dolomite; G—Granite; L—Limestone; S—Sandstone).

| Type of Rock | $UCS$/MPa | $E_{ni}$/MPa | $m$ | Coefficients of Determination $R^2$ |
|---|---|---|---|---|
| D01 | 62.15 | 5.82 | 0.9117 | 0.9951 |
| D02 | 115.06 | 11.01 | 0.8911 | 0.9977 |
| D03 | 124.10 | 12.79 | 0.9039 | 0.9792 |
| D04 | 150.59 | 13.48 | 0.9626 | 0.9935 |
| D05 | 136.74 | 14.38 | 0.8759 | 0.9917 |
| D06 | 176.04 | 16.30 | 0.9216 | 0.9907 |
| D07 | 168.37 | 17.41 | 0.9389 | 0.9922 |
| D08 | 189.53 | 19.17 | 0.8630 | 0.9817 |
| D09 | 192.21 | 19.94 | 0.9478 | 0.9976 |
| D10 | 185.34 | 20.28 | 0.8880 | 0.9927 |
| G01 | 36.24 | 3.14 | 0.6508 | 0.9939 |
| G02 | 49.95 | 5.57 | 0.7621 | 0.9840 |
| G03 | 69.71 | 6.74 | 0.8872 | 0.9859 |
| G04 | 70.76 | 8.28 | 0.7335 | 0.9998 |
| G05 | 84.64 | 9.97 | 0.7789 | 0.9934 |
| G06 | 108.23 | 11.70 | 0.8899 | 0.9977 |
| G07 | 140.88 | 13.81 | 0.9517 | 0.9861 |
| G08 | 153.39 | 14.42 | 0.9488 | 0.9758 |
| G09 | 171.11 | 17.03 | 0.9455 | 0.9846 |
| G10 | 202.02 | 19.76 | 0.9919 | 0.9951 |

**Table 1.** *Cont.*

| Type of Rock | UCS/MPa | $E_{ni}$/MPa | $m$ | Coefficients of Determination $R^2$ |
|---|---|---|---|---|
| L01 | 23.72 | 1.91 | 0.8868 | 0.9907 |
| L02 | 48.96 | 4.09 | 0.9927 | 0.9904 |
| L03 | 48.28 | 5.12 | 0.9867 | 0.9850 |
| L04 | 62.50 | 5.58 | 0.8569 | 0.9850 |
| L05 | 75.60 | 6.81 | 1.0282 | 0.9733 |
| L06 | 84.66 | 7.58 | 0.8631 | 0.9943 |
| L07 | 99.42 | 8.90 | 0.8684 | 0.9897 |
| L08 | 107.62 | 10.07 | 0.8406 | 0.9888 |
| L09 | 116.59 | 10.23 | 0.8039 | 0.9948 |
| L10 | 136.49 | 12.40 | 0.8981 | 0.9870 |
| S01 | 32.69 | 2.88 | 1.0102 | 0.9954 |
| S02 | 46.07 | 3.93 | 1.0529 | 0.9868 |
| S03 | 70.80 | 5.37 | 0.9518 | 0.9947 |
| S04 | 44.31 | 5.66 | 0.8245 | 0.9940 |
| S05 | 89.10 | 8.69 | 0.9203 | 0.9853 |
| S06 | 76.43 | 7.48 | 1.0095 | 0.9762 |
| S07 | 104.29 | 9.61 | 1.0199 | 0.9775 |
| S08 | 95.49 | 10.69 | 0.9901 | 0.9780 |
| S09 | 155.85 | 14.17 | 1.0273 | 0.9916 |
| S10 | 132.84 | 14.87 | 1.0529 | 0.9922 |

*5.3. The m Value Range of the Rock Material*

The range of variation in the *m* values for the different rock types was 0~2, as listed in Table 1. Based on the unified nonlinear elastic model, the experimental curves for the different rock types were fitted using the least square algorithm method when the *m* values were increased from 0.1 to 2.0 with a gradient of 0.1 and when the $E_{ni}$ values were not fixed. The scatter plots of the coefficients of correlation $R^2$ values and the *m* values are listed in Figure 11, and the scatter plots of the $E_{ni}$ and *m* values are listed in Figure 12. All the curves in Figure 11a–d are parabolic in their shape; that is, the $R^2$ values near the extreme point are high and the $R^2$ values far from the extreme point are low, which conforms to the local optimal feature of the least squares algorithm method. It is clear that the parameters fit to the unified nonlinear elastic model were optimal and unique. The highest points of the different curves in each picture are close, and the two side curves decrease rapidly in Figure 11a–d, indicating that the appropriate *m* values are in a small range and the $R^2$ values decrease rapidly when they are greater than this range.

The *m* values of different rock types were counted under the following three conditions: maximum $R^2$, an $R^2$ greater than 0.95 and an $R^2$ greater than 0.8. The maximum and minimum values of *m* were used to describe its range in each condition, and the results are listed in Table 2. The ranges of the *m* values were 0.65~1.05, 0.28~1.26 and 0.00~1.52 in the above three conditions for the $R^2$ values, the differences were small and the upper and lower limits were close. The fitting degree will be poor when the *m* values are greater than 1.52, and the unified nonlinear elastic model will not be applicable. The nonlinear elastic deformation characteristics of each rock type were different and are easy to ignore since their differences are small. In previous studies, there was no model or method that could accurately describe these behaviors, whereas the unified nonlinear elastic model can distinguish and accurately describe them.

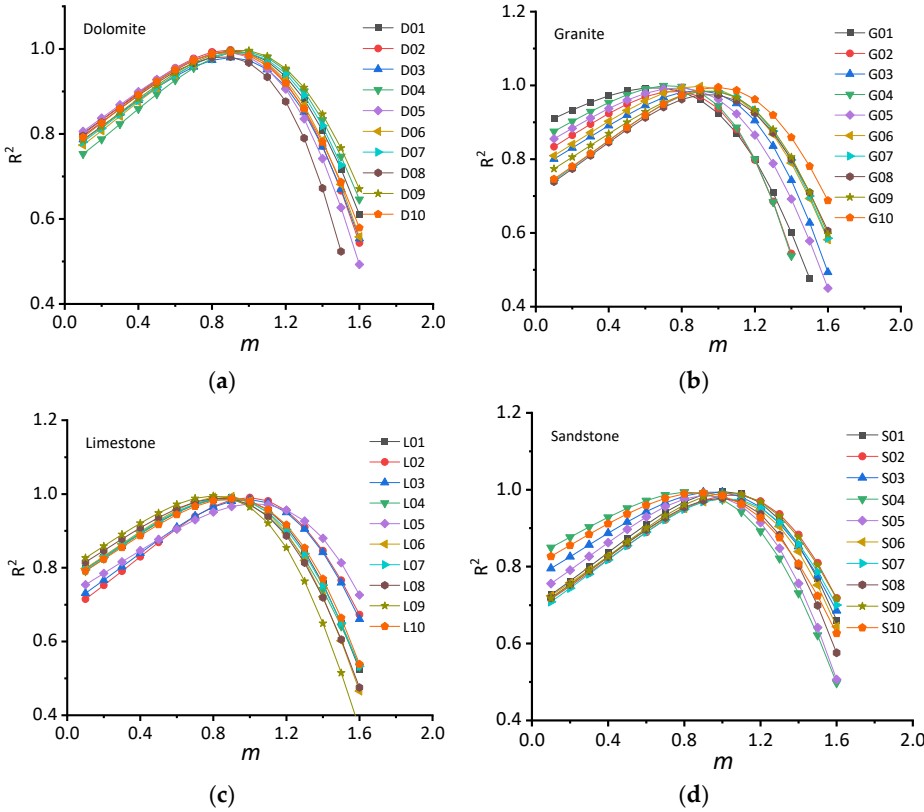

**Figure 11.** The $R^2$ values with different *m* values for different rock types. (**a**—Dolomite; **b**—Granite; **c**—Limestone; **d**—Sandstone).

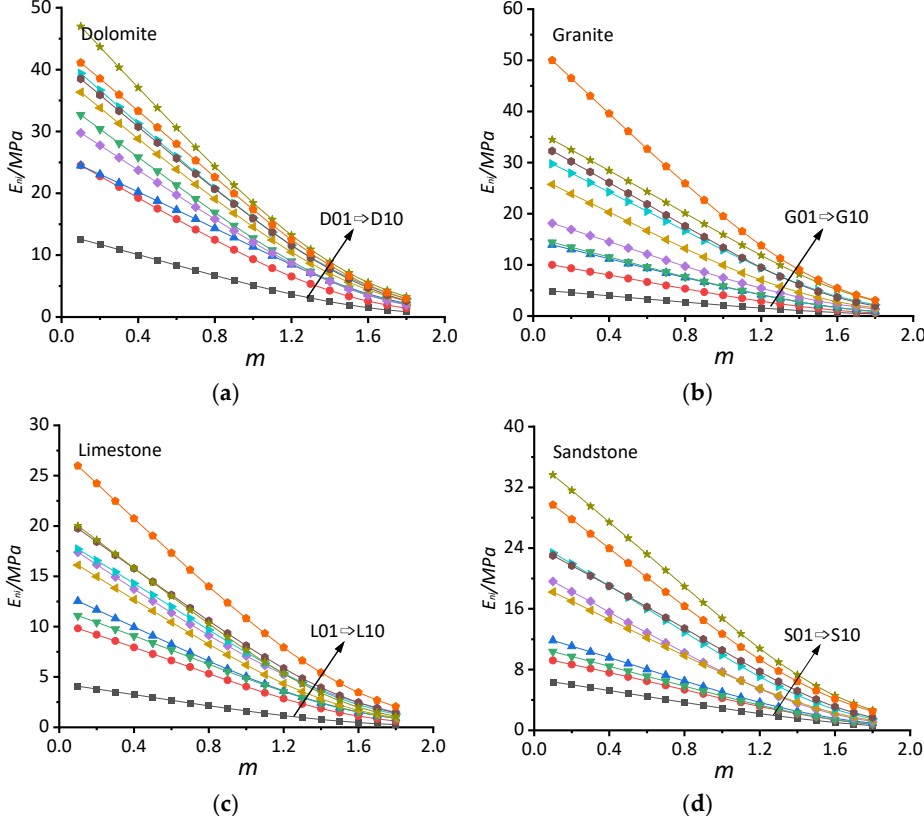

**Figure 12.** The $E_{ni}$ values with different *m* values for different rock types. (**a**—Dolomite; **b**—Granite; **c**—Limestone; **d**—Sandstone).

**Table 2.** Range of *m* values with different $R^2$ values.

| Type of Rock | Best-Fit Range | Range When $R^2$ Values Exceed 0.95 | Range When $R^2$ Values Exceed 0.80 |
|---|---|---|---|
| Dolomite | 0.86~0.96 | 0.58~1.20 | 0.08~1.46 |
| Granite | 0.65~0.99 | 0.28~1.23 | 0.00~1.47 |
| Limestone | 0.80~1.03 | 0.51~1.22 | 0.04~1.52 |
| Sandstone | 0.82~1.05 | 0.49~1.26 | 0.00~1.32 |

In addition, the *m* value corresponding to the simple exponential model was 1, and that corresponding to the *BB* model was 2, so the former can be used to describe the nonlinear elastic deformation of some rocks, whereas the latter has poor applicability.

*5.4. The Range of $E_{ni}$ Values for the Rock Material*

The scatter plots of the $E_{ni}$ and *m* values are listed in Figure 12. All the curves in Figure 12a–d show a downwards trend, indicating that the $E_{ni}$ values are negatively correlated with the *m* values, which follows the same rule of fit for a single curve shown in Figure 6. The $E_{ni}$ values increase as the serial numbers of the different rocks increases, and the position of the curve is also higher. The linear relationship of all the curves was good when the *m* values were less than 1.2, and the curves gradually approach each other with decreasing rates when the *m* values were greater than 1.2, as shown in Figure 12a–d. The scatter points hardly overlap with one another, that is, the $E_{ni}$ values have good independence and non-repeatability. The $E_{ni}$ values depend on the basic properties of the rocks, such as the *m* values, and the basic attributes are natural. It is not feasible to obtain the $E_{ni}$ values of all rocks by using the same *m* value, which is why the unified nonlinear elastic model contains both $E_{ni}$ and *m* values.

The range of $E_{ni}$ values for the different rock types were determined and are listed in Table 3. The range of $E_{ni}$ values of different rock types is slightly different, and dolomite and granite have larger ranges than limestone and sandstone. In general, the maximum $E_{ni}$ values for all the rock types were less than 21 MPa.

**Table 3.** The range of $E_{ni}$ values for different rock types.

| Type of Rock | Range of $E_{ni}$/MPa |
|---|---|
| Dolomite | 5.82~20.28 |
| Granite | 3.14~19.76 |
| Limestone | 1.91~12.40 |
| Sandstone | 2.88~14.87 |

## 6. Application of the New Model

According to the data in Table 1, the uniaxial compressive strength (*UCS*) values and the initial elastic modulus $E_{ni}$ values for the different rocks were counted, and the scatter diagrams are shown in Figure 13a–d. The results of the linear fitting of the data are also shown in these diagrams. The coefficients of correlation $R^2$ values for the four rock types are greater than 0.90, indicating that the *UCS* values and the $E_{ni}$ values have a good linear relationship [39]. The slope and intercept of different linear fittings and the $R^2$ values are listed in Table 4. Granite and limestone have the highest $R^2$ values, followed by dolomite and sandstone. The slopes of the linear formula for the different rocks were close, whereas the intercepts were different. When marble was added as another rock type, the scatter plots of the *UCS* and $E_{ni}$ values for the five rock types were determined and are listed in Figure 14. The results of the linear fitting of the data are also shown in Figure 14. In order to ascertain whether the experimental values were, in fact, unique to the present model, the 95% prediction band for the present model was determined and is represented by the red band. Almost all the experimental data fall within the 95% prediction band of the present

model, which means that the *UCS* values can be predicted by the $E_{ni}$ values. The linear fitting formula is listed below:

$$UCS = 0.927E_{ni} + 11.26 \tag{24}$$

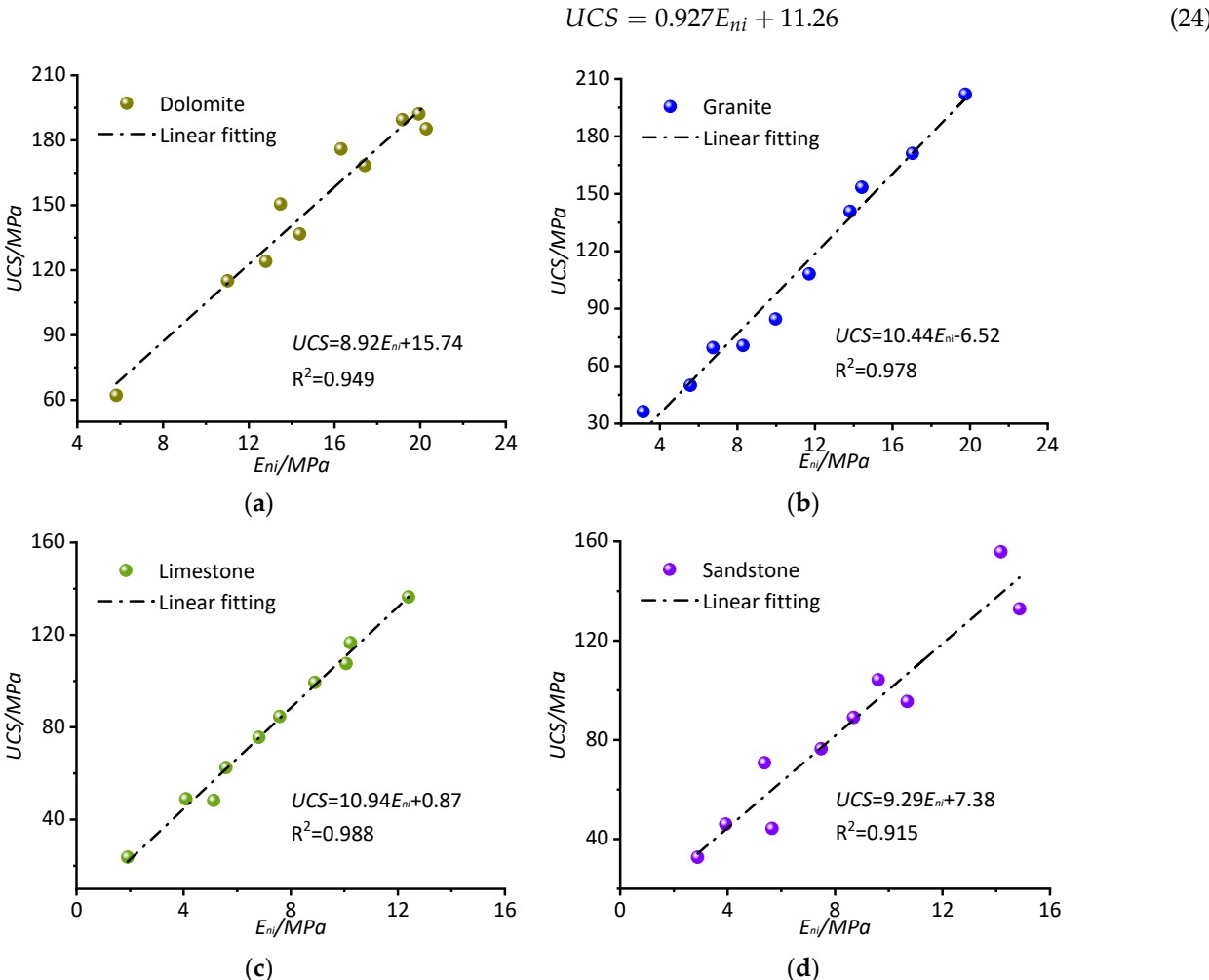

**Figure 13.** Correlations between *UCS* and $E_{ni}$ for different rock types. (**a**—Dolomite; **b**—Granite; **c**—Limestone; **d**—Sandstone).

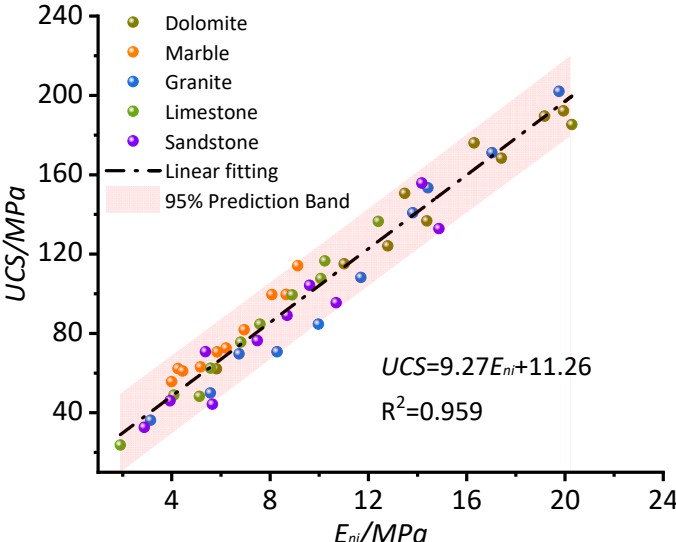

**Figure 14.** Comprehensive relationship between *UCS* and $E_{ni}$ for all the rock types.

**Table 4.** Linear relationships between *UCS* and $E_{ni}$ of different rock types.

| Type of Rock | Linear Fitting y = ax + b | | Coefficients of Determination $R^2$ |
|---|---|---|---|
| | a | b | |
| Dolomite | 8.92 | 15.74 | 0.950 |
| Granite | 10.44 | −6.52 | 0.978 |
| Limestone | 10.94 | 0.87 | 0.988 |
| Sandstone | 9.29 | 7.38 | 0.915 |

## 7. Summary and Conclusions

In this paper, the initial elastic modulus $E_{ni}$ and modulus change rate *m* are introduced in order to establish a new unified nonlinear elastic model for the nonlinear elastic deformation of intact rocks. Based on the RMT experimental system, a large number of uniaxial compression tests for dolomite, granite, limestone and sandstone were performed, and their nonlinear deformation stress-strain curves were employed to fit the unified nonlinear elastic model. The main outcomes are summarized as follows:

(1) The nonlinear elastic characteristic is obvious in the initial compression stage of the intact rock, which is mainly attributable to the porosity and/or microfractures present depending on the different rock types.

(2) In the unified nonlinear elastic model, the physical meanings of two key parameters, namely, the $E_{ni}$ representing the initial elastic modulus and the *m* representing its change rate, were clearly defined. $E_{ni}$ determines the slope in the initial stage of deformation, and *m* determines the degree of nonlinearity in the middle and later stages. The unified nonlinear elastic model not only covers the existing nonlinear elastic models, such as the simple exponential model (*m* = 1) and the BB model (*m* = 2) of the joint and the two-part Hooke's model (*m* = 1) of the rock, but is also able to provide some new models.

(3) The unified nonlinear elastic model was able to describe the nonlinear deformation elastic characteristics of the intact rocks in their initial compression stage rather well. This was justified as the data fit with the experimental results.

(4) The $E_{ni}$ values have a good linear correlation with the uniaxial compressive strength (*UCS*) values of the intact rocks. Based on this observation, a new empirical formula for the prediction of the *UCS* value using the $E_{ni}$ value was proposed.

(5) The unified nonlinear elastic model can also be employed to describe the nonlinear elastic deformation behavior of the jointed rock mass, but the range of the *m* values may be different from that of the intact rocks, which needs further investigation. For example, we could divide the jointed rock into two distinct portions (intact rock and joints) and analyze their deformation individually. Afterwards, they could be combined according to the principles of "strain superimposition" [40] or "energy equivalence" [41].

**Author Contributions:** Conceptualization, C.C. and S.C.; methodology, C.C.; software, H.L.; validation, Y.Z., H.L. and Y.W.; formal analysis, H.L.; investigation, Y.W.; data curation, Y.Z.; writing—original draft preparation, C.C.; writing—review and editing, S.C.; funding acquisition, C.C. All authors have read and agreed to the published version of the manuscript.

**Funding:** This research was funded by the National Natural Science Foundation of China under Grant 42277186 and 51927815.

**Institutional Review Board Statement:** Not applicable.

**Informed Consent Statement:** Not applicable.

**Data Availability Statement:** Not applicable.

**Conflicts of Interest:** The authors declare no conflict of interest.

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
