# Peer review of "A Unified Nonlinear Elastic Model for Rock Material"

_applsci, doi:10.3390/app122412725_

Round 1

Reviewer 1 Report

The paper concerns a stress-strain relation which takes inro account nonlinear features of the rock material. 

The explanation of the model is a bit unclear.

The following items should be addressed:

-p.5, Eq(1): where is a hyperbolic function there?

Above Eq. (4): proposed classical... (it might be called classical later by others), it is better to drop this word.

Numbering in Sec. 3 : 3.1 is used twice etc.

p.7, Fig.2, linear or nonlinear strains.  How it might be that  nonlinear strain can be less than the linear one? 

In particular, 1D expression for elastic medium v=u_x+0.5 u_x^2, and v>u_x).

The notion material length should be explained, is it introduced by the authors? 

Page 8, Section "A new..." is it already the authors contribution? 

It should be explaned more clearly. what is the novelty, why the modifications are of the form, say, Eq. (9).

English should be improved.

The paper needs in major revisions to be suitable for publication.

Reviewer 2 Report

Dear Authors the following corrections or amendements should be useful to ameliorate the paper. Thanks

1) Keywords: “joint” as keyword seems not to much discussed along the paper

2) Check number of chapters especially n.3

3) In 3.1: please comment the way to prepare a routine in order  to select the moduli for specific calculation or modelling in real problems  (e.g. a shallow tunnel, a pillar in a mine)

4) In.3.1: heterogeneous material means also composite materials: please refer to Del Greco et al.  International Journal of Rock Mechanics and Mining Sciences and1993, 30(7), pp. 1539–1543 to cite an interesting comparison

5) Also provide a short comment about differences with anisotropic rock material (gneiss)

6) In 3.3 please refer also to paper by Ocak Estimating the modulus of elasticity of the rock material from compressive strength and unit weight, October 2008, Journal- South African Institute of Mining and Metallurgy 108(10):621-626

7) And to

Davarpanah, M., Somodi, G., Kovács, L. et al. Experimental Determination of the Mechanical Properties and Deformation Constants of Mórágy Granitic Rock Formation (Hungary). Geotech Geol Eng 38, 3215–3229 (2020). https://doi.org/10.1007/s10706-020-01218-4

8) In 5.1 the uniaxial stress‒strain curves were recorded:

which technique has been used for strain measurements? Which was the precision?

9) How the specim behave in failure? Brittle or similar? Please comment in the captions of figures or result discussion

10) along the text, distinguish the “m” parameter with the “m” parameter used in Hoek criterion

11) Conclusions: add a comment on porosity and limits for anisotropic and composite rock materials

Round 2

Reviewer 1 Report

The authors properly responded to the comments, and now the paper is suitable for publication.

Reviewer 2 Report

Thanks for the amendments.

Take care that correct authorship of  ref.36 is

Del Greco O., Ferrero A.M., Oggeri C.

and that DOI is 10.1016/0148-9062(93)90153-5

Regards